# Excess morbidity and mortality among survivors of childhood acute lymphoblastic leukaemia: 25 years of follow-up from the United Kingdom Childhood Cancer Study (UKCCS) population-based matched cohort

Eleanor Kane ,[1] Sally Kinsey,[2] Audrey Bonaventure,[3] Tom Johnston,[1] Jill Simpson,[1] Debra Howell ,[1] Alexandra Smith ,[1] Eve Roman [1]

[1]Department of Health Sciences, University of York, York, UK
[2]Leeds Teaching Hospitals NHS Trust, Leeds, UK
[3]Epidemiology of Childhood and Adolescent Cancers Team, INSERM, Paris, France

**Correspondence to**
Professor Eve Roman;
eve.roman@york.ac.uk

## ABSTRACT

**Objectives** To examine morbidity and mortality among teenagers and young adults (TYAs) previously diagnosed with acute lymphoblastic leukaemia (ALL) in childhood, and compare to the general TYA population.

**Design** National population-based sex-matched and age-matched case-control study converted into a matched cohort, with follow-up linkage to administrative healthcare databases.

**Setting** The study population comprised all children (0–14 years) registered for primary care with the National Health Service (NHS) in England 1992–1996.

**Participants** 1082 5-year survivors of ALL diagnosed<15 years of age (1992–1996) and 2018 unaffected individuals; followed up to 15 March 2020.

**Main outcome measures** Associations with hospital activity, cancer and mortality were assessed using incidence rate ratios (IRR) and differences.

**Results** Mortality in the 5-year ALL survivor cohort was 20 times higher than in the comparison cohort (rate ratio 21.3, 95% CI 11.2 to 45.6), and cancer incidence 10 times higher (IRR 9.9 95% CI 4.1 to 29.1). Hospital activity was increased for many clinical specialties, the strongest associations being for endocrinology; outpatient IRR 36.7, 95% CI 17.3 to 93.4 and inpatient 19.7, 95% CI 7.9 to 63.2 for males, and 11.0, 95% CI 6.2 to 21.1 and 6.2 95% CI 3.1 to 13.5, respectively, for females. Notable excesses were also evident for cardiology, neurology, ophthalmology, respiratory medicine and general medicine. Males were also more likely to attend gastroenterology; ear, nose and throat; urology; and dermatology, while females were more likely to be seen in plastic surgery and less likely in midwifery.

**Conclusions** Adding to excess risks of death and cancer, survivors of childhood ALL experience excess outpatient and inpatient activity across their TYA years, which is not related to routine follow-up monitoring. Involving most clinical specialties, associations are striking, showing no signs of diminishing over time. Recognising that all survivors are potentially at risk of late treatment-associated effects, our findings underscore the need to

### Strengths and limitations of this study

► With minimal selection bias and loss to follow-up, findings from this national study are based on linked hospital data, not self-report.

► Ensuring homogeneity of treatment, diagnostic dates are contemporaneous with the Medical Research Council's national randomised controlled trial of ALL treatment (UKALL XI), within which 90.5% of those who survived were enrolled.

► The individually sex-matched and age-matched general population controls had a similar deprivation distribution to cases, enabling robust baseline effect measures to be calculated across the 25-year follow-up period.

► Analyses are constrained by the fact that hospital episode statistics are primarily collected for administrative and clinical purposes, and not for research.

take prior ALL diagnosis into account when interpreting seemingly unrelated symptoms later in life.

## INTRODUCTION

Comprising around 40% of all cancers diagnosed in children before 15 years of age, acute lymphoblastic leukaemia (ALL) is the most common paediatric malignancy in high-income countries. Epitomising one of the major therapeutic success stories of the last 50 years, 5-year overall survival for childhood ALL is now around 90% in high-income settings.[1] As a result, the global prevalence of survivors is increasing year on year, and will continue to do so as treatments improve, populations age, costs fall and new clinical collaborations between high-income and low-income countries are forged.[2]

In contrast to many other childhood cancers, ALL treatment is prolonged, with current protocols typically administering chemotherapy in phases extending over a 2-year to 3-year period.[3] That children who survive ALL tend to have higher mortality and morbidity in later life, including subsequent cancers and a wide range of other therapy-related conditions (table 1), is established.[4–7] With respect to the latter, however, much of the information on late effects among survivors has come either from analyses of hospitalisations or from questionnaires completed by consenting individuals.[8–17] As is evident from the studies summarised in table 1 (which includes those that examined all leukaemias combined and those that also included young adults), less attention has been paid to chronic conditions that do not require hospital admission,[7 18–20] and to morbidity patterns that change over time.[7 12 21]

The longitudinal data examined in this report contain information on hospital outpatient attendances, as well as inpatient episodes, cancer registrations and deaths. Adding significantly to the existing knowledge base, a large cohort of children (<15 years) diagnosed with ALL in England during 1992–1996 and their individually sex-matched and age-matched population controls have been tracked for up to 25 years through their teenage and young adult (TYA) years. Increasing the power to detect effects should they exist, this diagnostic time period (1992–6) coincides with the UKALL XI randomised controlled trial (RCT)[22]; the last very intensive UK RCT prior to the era of clinical investigations enabling reduction in therapy for patients with rapid early response to treatment.[23]

## METHODS

Data are from the United Kingdom Childhood Cancer Study (UKCCS); a national population-based case-control study established in the 1990s to investigate potential causes of childhood cancer that has now been converted to a population-based cohort study to examine survivorship.[24] Full details of the UKCCS's original methods and ethical permissions have been published.[25–27] Briefly, the study population comprised all children (0–14 years) registered for primary care with the National Health Service (NHS) 1992–1996 (England and Wales) and 1991–1995 (Scotland). All children newly diagnosed with cancer (cases) were ascertained via proactive notification systems established in all treatment centres across England, Scotland and Wales; and each child whose parents agreed to be interviewed (87%; all cancers combined) was individually matched on sex, date of birth and region of residence to up to 10 randomly selected controls. The general practitioners (GPs) of the first two controls identified ('first-choice controls') were approached and, with their permission, the parents of the children were contacted and asked to participate in the study. Overall, 72% of first-choice control families participated; but if the GP refused permission to contact the parents, or the parents themselves declined, the next control on the list was selected,

and so on until two control families agreed.[25–27] For methodological purposes,[26 27] details of all non-participating cases and controls were retained in the UKCCS registration database.

With a view to examining childhood cancer sequelae, the UKCCS in England now operates on a legal basis that permits information to be obtained from NHS administrative healthcare records without explicit consent; and all registered cases and first-choice controls, along with replacement controls whose parents participated, are now being prospectively tracked by NHS Digital (https://digital.nhs.uk/) via linkage to nationwide (England) information on deaths, cancer registrations, and Hospital Episode Statistics (HES; inpatient day and overnight admissions, and outpatient appointments and visits). Mortality, cancer incidence and secondary healthcare utilisation of children diagnosed with ALL who survived for 5 years or more is compared here to that of individually sex-matched and age-matched general-population controls using a matched population-based cohort design.

### Data and statistical analysis

The number of ALL cases diagnosed in the original study and their corresponding first-choice controls, as well as subsequent deaths and losses to follow-up occurring in England within 5 years of ALL diagnosis/pseudodiagnosis are shown in figure 1. Overall, 1574/1580 (99.6%) cases and 2918/2923 (99.8%) of their first-choice controls were successfully traced and linked to national administrative databases. During the study period (1992–1996), 1372/1580 children were diagnosed with ALL in England, and the parents of 92% (1262/1372) of these agreed to participate, each of whom had two sex-matched and age-matched first-choice controls selected (n=2524).

In the original case-control study, each unaffected control child was assigned a pseudodiagnosis date that corresponded to their matched case's exact age at diagnosis; and the dates of diagnosis/pseudo-diagnosis now mark the start of follow-up for the matched cohorts. Of the 1104 ALL cases (80.5%) diagnosed in England who survived for 5 years or more, 1004 (90.9%) had B-ALL and 70 (6.3%) T-ALL. The cell lineage of the remaining eight ALL survivors is unknown; they were not entered into a clinical trial and the parents did not participate in the original study (hence no controls were selected). The 22 surviving case children with Down syndrome (no first-choice control had Down syndrome) and their matched controls were excluded from the present survivorship analysis because Down syndrome is a well-established risk factor for ALL, and it is also associated with a range of other morbidities.

For all of the 1082 cases and 2018 controls included in this report (figure 1), linked healthcare data were available on cancer registrations and deaths up to March 2020, inpatient HES from April 1997 to March 2019, and outpatient HES from April 2003 to March 2019. Follow-up ended either at the date of death, 25 years post diagnosis, or end date for hospital data. For controls,

**Table 1** Studies reporting morbidity among 5-year survivors of leukaemia diagnosed in children, teenagers and young adults: both acute lymphoblastic leukaemia (ALL) alone, and all leukaemias combined

| Study | 5-year survivors | | | Follow-up* source, timing post diagnosis | Outcome measures | Main findings |
|---|---|---|---|---|---|---|
| | Setting, timeframe | Numbers, completeness | Comparator source | | | |
| **ALL** | | | | | | |
| Childhood Cancer Survivor Study, CCSS[8] | USA and Canada: 31 centres 1970–1999 | 6148<21 years ~72% consented at baseline | 5051 controls Closest aged siblings in random sample of all childhood cancer survivors | Biannual questionnaires 2000–2017 5–46 years post diagnosis (median 22 years) | Conditions considered severe by the respondents, distributed by treatment era and treatment group | 11% of survivors diagnosed in the 1990s receiving standard treatment reported at least one severe chronic condition in the 20 year after diagnosis; rate ratio 1.9 (95% CI 1.5 to 2.3) compared with controls. RRs>2 for stroke, major joint replacement, and diabetes |
| Adult Life after Childhood Cancer in Scandinavia, ALiCCS†[9] | Denmark, Sweden, Iceland & Finland: National cancer registries 1970–2008† | 3391<20 years 100% traced | 129 828 controls matched on sex, age, and country National population registers | National hospital registers 1975–2012 5–42 years post diagnosis (median 16 years) | First inpatient admission for 120 conditions, total and by organ system | Survivors were nearly twice (rate ratio=1.95, 95% CI 1.83 to 2.07) as likely to be hospitalised for ≥1 of 120 conditions than controls. RRs>2 for diseases of the endocrine, nervous and circulatory systems; infections; blood/blood organs, second cancers and benign neoplasms |
| St Jude Lifetime Cohort, SJLIFE[12] | USA: St Jude Children's Research Hospital 1963–2003 | 980<18 years alive 10 years after diagnosis ~67% consented | 272 controls Friends, non-first-degree relatives of hospital patients, hospital employees and volunteers | In-person interview 2015 12–52 years post diagnosis (median 25 years) | Conditions considered by respondents to be moderate/severe, overall and by organ system | By age 30, survivors reported 3.2 (95% CI 2.9 to 3.4) chronic conditions compared with 1.2 (95% CI 1.0 to 1.4) among controls. Conditions varied by treatment era, with endocrine and musculoskeletal conditions contributing more to the cumulative burden in the 1990s and 2000s than in earlier decades |

Continued

**Table 1** Continued

| Study | 5-year survivors | | | | | |
| | Setting, timeframe | Numbers, completeness | Comparator source | Follow-up* source, timing post diagnosis | Outcome measures | Main findings |
| --- | --- | --- | --- | --- | --- | --- |
| Nordic Society of Paediatric Haematology and Oncology ALL registry, NOPHO[7] | Denmark: ALL national registry 1994–2015 | 675 aged 1–<18 years alive 2.5 years after diagnosis ~96.4% traced | 6750 controls matched on age, sex and region National population register | National primary care register 1997–2018 and national hospital registers 1997–2017 2.5–24 years post diagnosis (median 13.5 years) | Rates of primary care and hospital (in/outpatient) contacts, by year since diagnosis (2.5–17.5 years post-diagnosis) | At 2–3 years post diagnosis, survivors had 14.2 hospital contacts (95% CI 13.4 to 15.1), 30 times more than controls (incidence rate ratio=31.5, 95% CI 28.3 to 35.1); contacts decreased over time but remained above baseline through to end of follow-up (IRRs≥2). For primary care, survivors started with 4.75 (95%CI 4.41 to 5.11) daytime contacts, nearly twice that among controls (IRR=1.85, 95% CI 1.71 to 2.00); contacts remained at 3–5 times/yr but IRRs declined and were not statistically significant >12 years post diagnosis. |
| Swiss Childhood Cancer Survivor Study, SCCSS[17] | Switzerland: Swiss Childhood Cancer Registry 1976–2005 | 511<16 years, alive in 2007 and aged ≥16 years at questionnaire 72% participated | 702 siblings | Questionnaire 2007–2012 5–36 years post-diagnosis | Self-reported cardiovascular diseases | 14% of survivors reported having at least one cardiovascular disease, nearly twice that of their siblings (OR=1.9, 95% CI 1.3 to 2.8) |
| Childhood, adolescent and young adult cancer survivors, CAYACS[19] | Canada: British Columbia cancer registry‡ 1970–1992 | 242<20 years alive at 31 December 2000 ~64% traced | 11 570 controls frequency-matched on age and sex Health insurance plan | Health insurance claims 1998–2000 8–20 years post diagnosis | Physician visits over 3 year period excluding oncology, overall and by specialty | Survivors were nearly twice as likely as controls to visit a doctor at least once (rate ratio=1.95, 95% CI 1.8 to 2.1). RRs≥2 were found for ≥1 visit to neurology, ophthalmology, otolaryngology, internal medicine, surgery, urology, psychiatry and dermatology |
| Childhood Cancer Register of Southern Sweden, BORISS[18] | Sweden: Regional cancer registry, 1970–1999 | 213<18 years ~100% traced | 10 650 controls matched on sex, year of birth and county National population register | National hospital registers inpatient: 1975–2009, outpatient: 2001–2009 5–33 years post diagnosis (median 16 years) | Hospital contacts (in/outpatient), overall and for 20 disease categories | Survivors were over twice as likely to have ≥1 hospital contact than controls (OR=2.5, 95% CI 1.5 to 4.3), and were more than twice as likely to consult for disorders of the blood, endocrine system, nervous system, skin, infections or benign neoplasms |

Continued

**Table 1** Continued

| Study | 5-year survivors Setting, timeframe | Numbers, completeness | Comparator source | Follow-up* source, timing post diagnosis | Outcome measures | Main findings |
|---|---|---|---|---|---|---|
| University of Utah-Intermountain Healthcare[14] | USA: Primary Children's Hospital, Salt Lake City 1998–2008 | 176<22 years ~79% traced | 503 controls matched on sex and year of birth State population register | State hospital registers 2003–2013 5–10 years post diagnosis (median 9 years) | Rate of inpatient admissions over a maximum of 5 years | In the 5–10 years after diagnosis, the hospitalisation rate of survivors was 3.76 per 100 person years (95% CI 2.22 to 6.36), twice the rate among controls (relative hospitalisation rate=2.01, 95% CI 1.00 to 4.03) |
| *All leukaemias combined* | | | | | | |
| British Childhood Cancer Survivor Study, BCCSS[10] | UK: National Registry of Childhood Tumours 1940–1991 | 3544<15 years, alive in April 1997 ~87% being followed | Hospitalisation rates for cerebrovascular disease National population and hospital registers | National hospital registers 1997–2012 5–72 years post diagnosis | Cerebrovascular disease recorded in hospital discharge summaries | Survivors were almost five times more likely to have been hospitalised for cerebrovascular disease than expected (Age Standardised Hospitalisation Ratio=4.7, 95% CI 3.6 to 6.1) |
| Dutch Childhood Oncology Group-LATER Study[11] | Netherlands: paediatric oncology or haematology centres 1963–2001 | 1900<18 years, alive in January 1995 ~92% traced | 38 000 controls matched on sex and year of birth National population register | National hospital registers 1995–2015 5–52 years post diagnosis | Rate of inpatient admissions over a maximum of 10 years | The hospitalisation rate of survivors was almost twice that of controls (relative hospitalisation rate=1.87, 95% CI 1.65 to 2.12) |
| Leucémies de l'Enfant et l'Adolescent Cohort, LEA[16] | France: 16 cancer centres 1980–2012 | 1025<18 years, alive in 2007 and aged ≥18 years at last LEA medical examination 72% participated | 3203 controls matched on sex and age Volunteers for free medical examinations | Medical examinations at 2-year intervals until 10 years post diagnosis (or age 20), thereafter every 4 years 2–34 years post diagnosis (mean 16 years) | Metabolic syndrome found at medical examination | Prevalence of metabolic syndrome among survivors was 10.3%, more than twice that among controls (OR=2.49, 95% CI 1.91 to 2.35) |
| Scottish Cancer Registry[15] | UK: Scottish cancer registry 1981–2003 | 884<25 years ~100% traced | Hospitalisation rates National population and hospital registers | National hospital registers 1986–2009 5–28 years post diagnosis (mean 16 years) | First inpatient admission for particular diseases, overall, and for selected disease categories | Survivors were almost four times more likely to have been hospitalised than expected (Standardised Hospitalisation Ratio=3.9, 95% CI 3.6 to 4.3). SHR≥2 for diseases of the endocrine, nervous, circulatory, respiratory and digestive systems; infections and neoplasms |

Continued

**Table 1** Continued

| Study | 5-year survivors | | Comparator source | Follow-up* source, timing post diagnosis | Outcome measures | Main findings |
|---|---|---|---|---|---|---|
| | Setting, timeframe | Numbers, completeness | | | | |
| Washington State Cancer Registry[13] | USA: Washington State cancer registry 1982–2008 | 815<20 years Not evaluable | 8150 controls matched on sex and year of birth and alive at case's diagnosis State birth records | State hospital registers 1987–2013 5–27 years post diagnosis (median 9 years) | Rates of inpatient admissions, and hospitalisation or death for selected disease categories | The survivors' hospitalisation rate was almost three times that of controls (HR=2.8, 95% CI 2.2 to 3.5). HRs≥2 were observed for hospitalisation or death from endocrine, nervous, circulatory, respiratory, digestive, genitourinary, skin and musculoskeletal systems; blood diseases, infections and cancers |
| Clinical Practice Research Datalink[20] | UK: Clinical Practice Research Datalink 1998–2020 | 403≤25 years Not evaluable | 13 517 controls matched to all cancers on sex, year of birth, socioeconomic status and general practice Primary care registers | Primary care electronic health records linked to national hospital registers from age 18 years to 2020 5–22 years post diagnosis | Mean cumulative counts of primary care visits and hospital admissions for 183 conditions between ages of 18 and 35 or 45, overall and by organ system | Survivors' mean count of primary care visits was higher than among controls between ages of 18–35 years (23.5, 95% CI 19.8 to 29.3 compared with 4.0, 95% CI 3.9 to 4.1), and 18–45 years (29.8, 95% CI 24.7 to 36.0 compared with 7.2, 95% CI 7.1 to 7.4); as were their mean counts of admissions (18–35 years: 6.0, 95% CI 5.6 to 6.3 vs 1.1, 95% CI 1.0 to 1.1; 18–45 years: 7.9, 95% CI 7.0 to 8.0 vs 1.8, 95% CI 1.8 to 1.9). Survivors' cumulative burdens of cardiovascular conditions and endocrinopathies were higher than controls'. |

*Minimum to maximum possible follow-up relative to diagnosis date.
†Finland 1969–2012; Denmark 1977–2010; Sweden 1987–2009 with some regions starting in 1964–1987; Iceland 1999–2008.
‡CAYACS reported outcomes for all childhood cancer survivors, of which 20% had AL.
§Completeness could not be evaluated in the Washington State Cancer Registry Study as study subjects were not linked to a population register.

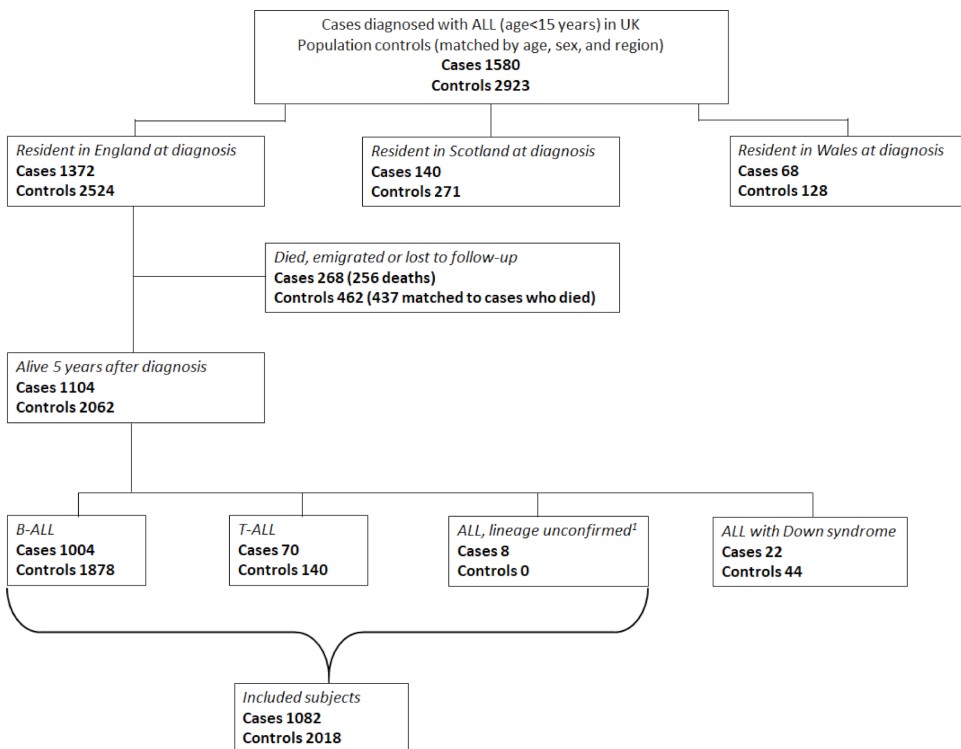

**Figure 1** Flow chart showing subjects targeted in the original United Kingdom Childhood Cancer case-control study and those included in the present cohort.

follow-up could also end when their matched case died. Migrations out of England and any subsequent returns were accounted for, and since there is no risk of a hospital visit during an admission, days spent in hospital were also excluded from the follow-up time.

Although the International Classification of Diseases (ICD) is used to code deaths and cancers, it is not routinely used in outpatient HES, and the ICD codes in inpatient HES are not easily assigned to specific events. Each HES episode (inpatient and outpatient) is, however, allocated to an individual consultant whose clinical specialty categorisation is recorded; and these categorizations were

used in the HES analyses presented here. First, since childhood ALL survivors are routinely monitored via outpatient visits to paediatrics, haematology and/or oncology in the UK, HES attendances to these clinical specialties were separated from those to 'other specialties' at the outset (figure 2). All subsequent specialty-specific analyses presented in this report were restricted to the 'other specialties'. As in previous reports,[28 29] a single inpatient admission and/or two or more face-to-face outpatient attendances were used to indicate potential health issues. Exact time-to-event methods were used to plot estimates of the hospital visit hazard rates before

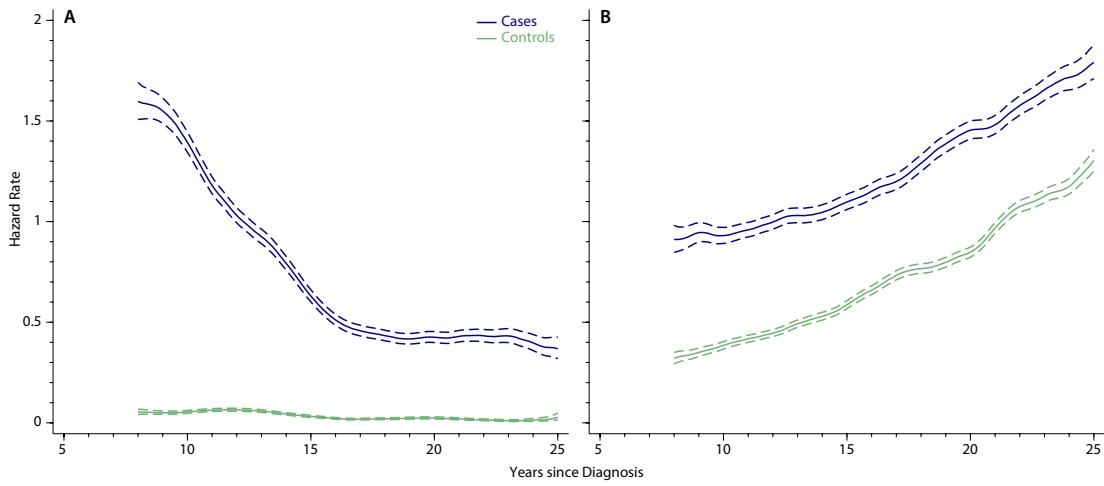

**Figure 2** Outpatient activity hazard rates per year and 95% CIs (dotted lines) for childhood (<15 years) acute lymphoblastic leukaemia 5-year survivors and their matched controls: (A) paediatrics, haematology and oncology, (B) all other clinical specialties.

calculating cumulative incidence (considering death as a competing risk), and risks (rate ratios and attributable risks) associated with specific clinical specialties. Hospital visit hazard rates (all visits) are estimated as the number of visits per person year while incidence rates of ever visiting specific clinical specialties are calculated per 1000 person years; for the latter, follow-up time was truncated at the first hospital visit to the specialty. All specialties were analysed, and findings for the 15 specialties with the highest cumulative incidences among the cases are presented. To analyse this prospective matched cohort study which included virtually all cases nationally, methods analogous to those employed in other studies of a similar design were used.[9 28 29] Importantly, cases and their randomly sampled controls were identical with respect to sex, age and socioeconomic status at baseline,[27] and remained so 5 years from diagnosis. With age, sex and socioeconomic status effectively controlled for in the study design, adjusting in the analyses will not change the risk estimates, and as such, rates, rate ratios and attributable risks derived using exact methods are presented without any adjustments. All analyses were conducted using Stata V.16.1.

### Patient public involvement and engagement (PPIE)

The original case-control study was established over 30 years ago, and did not benefit from formal links with patient/user groups. Members of the Children's Cancer and Leukaemia Group (CCLG, previously the UK Children's Cancer Study Group) were, however, fully involved throughout. More recently with respect to the present research, the acceptability of using patient data without explicit consent (as is the case in this report) has been positively discussed with patient groups and clinicians; and the study website (www.UKCCS. org) contains information about ongoing research activities and fair processing. It is our intention to establish a PPIE group to input into future research when the COVID-19 pandemic permits.

### RESULTS

The characteristics of individuals in the original case-control study and in the 5-year survivor cohort are shown in table 2. As expected, the 290 (21%) children with ALL who died within 5 years of diagnosis were more likely to have had T-cell disease ($\chi^2$=44.0, p<0.001), have been infants or older children ($\chi^2$=100.9, p<0.001), and were less likely to have participated in a clinical trial ($\chi^2$=11.6, p=0.001). Nevertheless the 5-year survivors and their controls remained similar with respect to sex ($\chi^2$=0.0002, p=0.99) and age ($\chi^2$=0.35, p=0.95) as well as deprivation (Index of Multiple Deprivation (IMD); $\chi^2$=0.32, p=0.57).

Tracking children from 5 years after ALL diagnosis to death or 25 years from diagnosis, this report's primary focus is TYA survivors who were aged around 9 years at the start of follow-up (median age 9.4 years, IQR 8.0–12.1 years) and almost 30 years by the end (median attained age 29.6 years, IQR 27.3–32.1). Between these ages, again as expected, 5-year ALL survivors continued

to experience more deaths (cumulative mortality 10.7%, 95% CI 9.0 to 12.8) than their general population counterparts (cumulative mortality 0.6%, 95% CI 0.3 to 1.1), yielding a mortality rate ratio of 21.3 (95% CI 11.2 to 45.6); and were more likely to have a subsequent cancer registration (non-ALL malignancy incidence rate ratio (IRR) 9.9, 95% CI 4.1 to 29.1), although the numbers are small (32 vs 6).

Childhood ALL survivors are regularly monitored throughout their TYA years either in paediatrics, haematology or oncology, as is evidenced in figure 2A which shows the outpatient activity of 5-year ALL survivors and their corresponding controls for these three clinical specialties combined (specialty-specific outpatient and inpatient data are shown in online supplemental figure 1). At around one visit per person in 20 years, hospital activity among population controls is very low, and varies little over time. By contrast, for ALL survivors, 8 years after diagnosis outpatient activity is around 35 times higher than the background rate, falling 10 years later to around 0.4 visits per case per year (figure 2A). In all other clinical specialties combined, outpatient activity among childhood ALL survivors is also consistently higher than expected; hazard rates in both cohorts increasing over time at the same steady rate (figure 2B).

More information on hospital activity is presented for the top 15 specialties (excluding paediatrics, haematology and oncology) for males and females separately in figure 3 (outpatients) and figure 4 (inpatients). In all figures, cumulative incidence frequencies are on the left and incidence rates on the right, with the ordering of specialties determined by the magnitude of the IRR. Most specialties are associated with high relative and attributable risks (inpatient and outpatient). Furthermore, although the magnitude of the estimates often varies between males and females, the overall pattern is broadly similar. In both sexes, for example, the strongest effects are seen for endocrinology; the IRRs for outpatients and inpatients being 36.7 (95% CI 17.3 to 93.4) and 19.7 (95% CI 7.9 to 63.2), respectively, for males, and 11.0 (95% CI 6.2 to 21.1) and 6.2 (95% CI 3.1 to 13.5) for females. Notable excesses (relative and attributable) are also evident for cardiology, neurology, ophthalmology, respiratory medicine and general medicine, for both genders. Male survivors were also more likely than their peers to attend gastroenterology; ear, nose and throat (ENT); urology; and dermatology, while female survivors were twice as likely as their controls to be referred to plastic surgery. Interestingly, no case-control differences are evident for trauma and orthopaedics, which has the highest background cumulative incidence in males. Likewise, among females, no differences were detected for obstetrics and gynaecology; however, it is notable that activity is lower than expected in midwifery (outpatient IRR 0.9, 95% CI 0.7 to 1.2; inpatient IRR 0.6, 95% CI 0.4 to 0.9).

Detailed inspection of HES data revealed no evidence of specialty-specific changes in either relative or attributable risks over the 20 year time frame (online supplemental figure 2); possible exceptions being falls in ranking for

**Table 2** Characteristics of acute lymphoblastic leukaemia (ALL) cases diagnosed <15 years in England 1992–96 and their age-matched and sex-matched first-choice controls*: total registered in the original case-control study, and 5-year survivors included in the present study

| | Original study | | 5 year survivors | |
| --- | --- | --- | --- | --- |
| | Cases, N (%) | Controls*, N (%) | Cases, N (%) | Controls*, N (%) |
| Total | 1372 (100) | 2524 (100) | 1082 (100) | 2018 (100) |
| Sex | | | | |
| Male | 763 (55.6) | 1410 (55.9) | 589 (54.4) | 1099 (54.5) |
| Female | 609 (44.4) | 1114 (44.1) | 493 (45.6) | 919 (45.5) |
| Lineage* | – | – | | |
| B-ALL | 1234 (89.9) | – | 1004 (92.8) | – |
| T-ALL | 120 (8.7) | – | 70 (6.5) | – |
| Age (years) at: | | | | |
| Diagnosis | | | | |
| <1 | 52 (3.8) | 86 (3.4) | 22 (2.0) | 41 (2.0) |
| 1–4 | 839 (61.2) | 1578 (62.5) | 703 (65.0) | 1331 (66.0) |
| 5–9 | 267 (19.5) | 484 (19.2) | 220 (20.3) | 402 (19.9) |
| 10–14 | 214 (15.6) | 376 (14.9) | 137 (12.7) | 244 (12.1) |
| Median (IQR) | 4.5 (3.0–7.6) | 4.4 (3.0–7.5) | 4.4 (3.0–7.1) | 4.3 (3.0–7.0) |
| Beginning of follow-up | | | | |
| Median (IQR) | – | – | 9.4 (8.0–12.1) | 9.3 (8.0–12.0) |
| End of follow-up† | | | | |
| Median (IQR) | – | – | 29.6 (27.3–32.1) | 29.5 (27.0–31.9) |
| IMD‡ quintile at diagnosis | | | | |
| Least deprived (1-3) | 834 (60.8) | 1533 (60.7) | 675 (62.4) | 1237 (61.3) |
| Most deprived (4-5) | 520 (37.9) | 975 (38.6) | 400 (37.0) | 766 (38.0) |
| Trial entry | | | | |
| Yes | 1215 (88.6) | – | 979 (90.5) | – |
| No | 157 (11.4) | – | 103 (9.5) | – |
| Person-years in follow-up† | – | – | 19 425.6 | 35 966.0 |
| Deaths in follow-up | – | – | 115 | 10 |
| Cumulative mortality, % (95% CI) | | | 10.7 (9.0 to 12.8) | 0.6 (0.3 to 1.1) |
| Mortality rate ratio (95% CI) | | | 21.3 (11.2 to 45.6) | |
| Cancer registrations§ in follow-up | – | – | | |
| Total | | | 60 | 23 |
| Cumulative incidence, % (95% CI) | | | 5.8 (4.5 to 7.4) | 1.4 (0.9 to 2.0) |
| Incidence rate ratio (95% CI) | | | 4.9 (3.0 to 8.9) | |
| Malignancies only | | | 32 | 6 |
| Cumulative incidence, % (95% CI) | | | 3.1 (2.2 to 4.3) | 0.4 (0.1 to 0.8) |
| Incidence rate ratio (95% CI) | | | 9.9 (4.1 to 29.1) | |

*Lineage was not confirmed for 18 cases, 8 of whom were alive 5 years after diagnosis.
†Follow-up time among controls was censored at the matched survivor's death if the survivor died during follow-up.
‡IMD: Index of Multiple Deprivation income domain, IMD was missing for 18 cases and 16 controls.
§Total cancer, including tumours that were registered as benign, in situ and of uncertain behaviour: ICD-10 codes C00-C90, C92-C97 and D00-D48. Malignancies only: C00-C43, C45-C90, C92-C97 (NB registrations for lymphoid leukaemia (C91) were excluded). Cumulative incidence of cancer was calculated considering death as a competing risk.

ophthalmology among males and dermatology among females. Furthermore, no differences between B-cell and T-cell survivors with respect to any of the outcomes were evident, although the number with T-cell disease is comparatively small (n=70).

## DISCUSSION

Including data on more than 1000 survivors of childhood ALL and twice as many age-matched and sex-matched general population controls, this national record-linkage study found that survivors continued to experience large

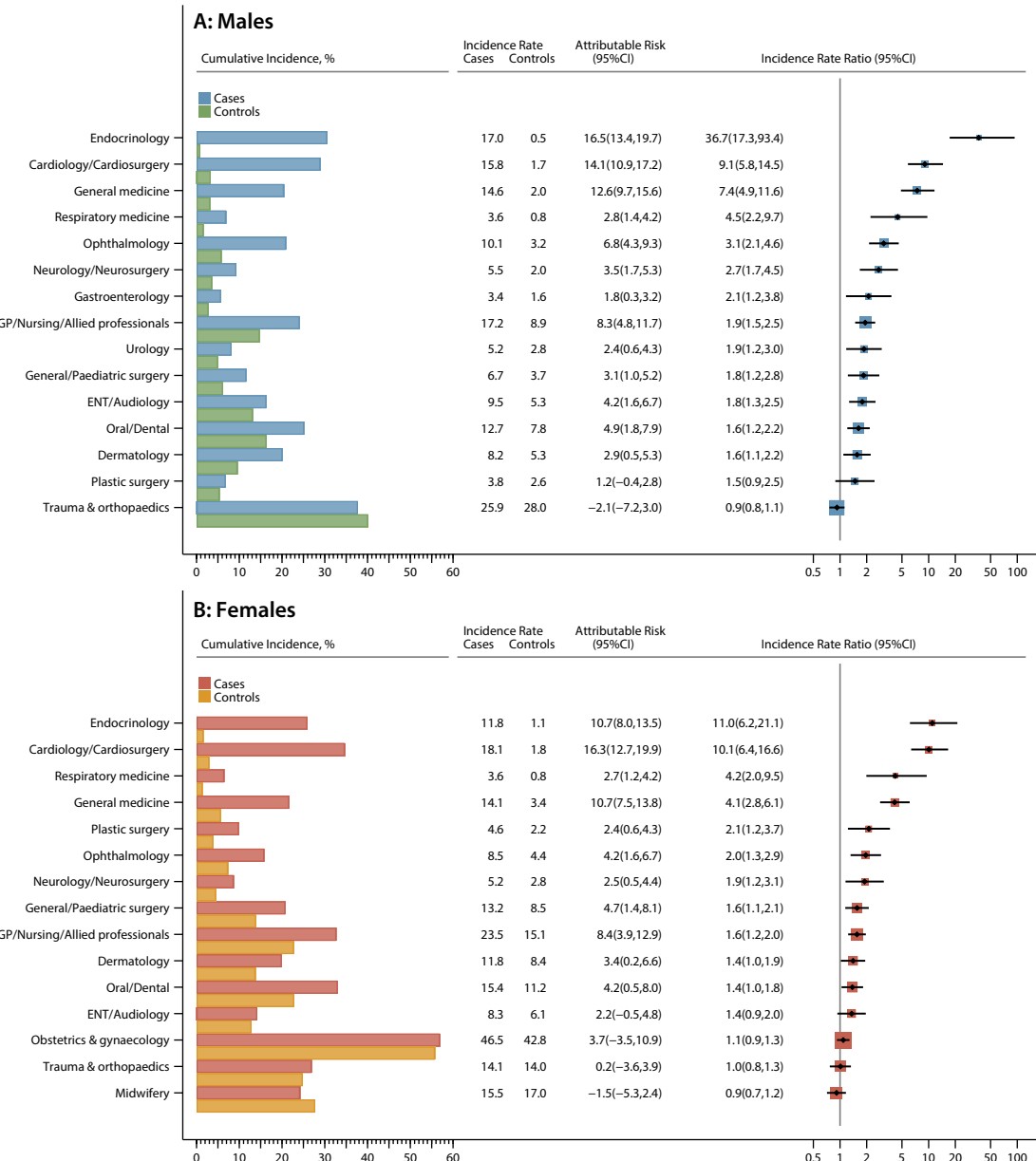

**Figure 3** Cumulative incidence (%), incidence rates (per 1000 person years), attributable risks, and incidence rate ratios for the top 15 outpatient specialties (excluding paediatrics, haematology and oncology) with two or more face-to-face visits in the 5–25 years following diagnosis (cases diagnosed <15 years, 1992–1996) and their matched population controls.

excesses in mortality and morbidity throughout their TYA years. With hospital activity being higher than expected in specialties covering most organ and tissue systems, survivors were more than twice as likely to fall under the care of endocrinology, cardiology, respiratory medicine, ophthalmology, neurology and/or gastroenterology. Moreover, aside from a gradual decline in visits to clinical specialties involved in follow-up monitoring (paediatrics, haematology and oncology), there was little indication of activity returning to general population levels up to 25 years after the initial ALL diagnosis. Among females, it is also notable that midwifery outpatient attendances were not increased, with admissions being lower than expected.

Our study, which uses linked rather than self-reported data and population controls, is one of only two[18] to separate secondary care activity associated with cancer follow-up from activity associated with other morbidities; finding that survivors not only experienced higher rates of admission but also higher rates of outpatient visits, neither of which abated over their TYA years. Reported less often are the types of conditions underlying these associations, with four studies assessing risks by organ system.[9 13 15 18] Survivors in these studies were diagnosed over a wider period than the study reported here, and had median follow-up times that were shorter; 9–17 years from diagnosis, compared with 25 years here. These studies found that survivors were at increased risk of

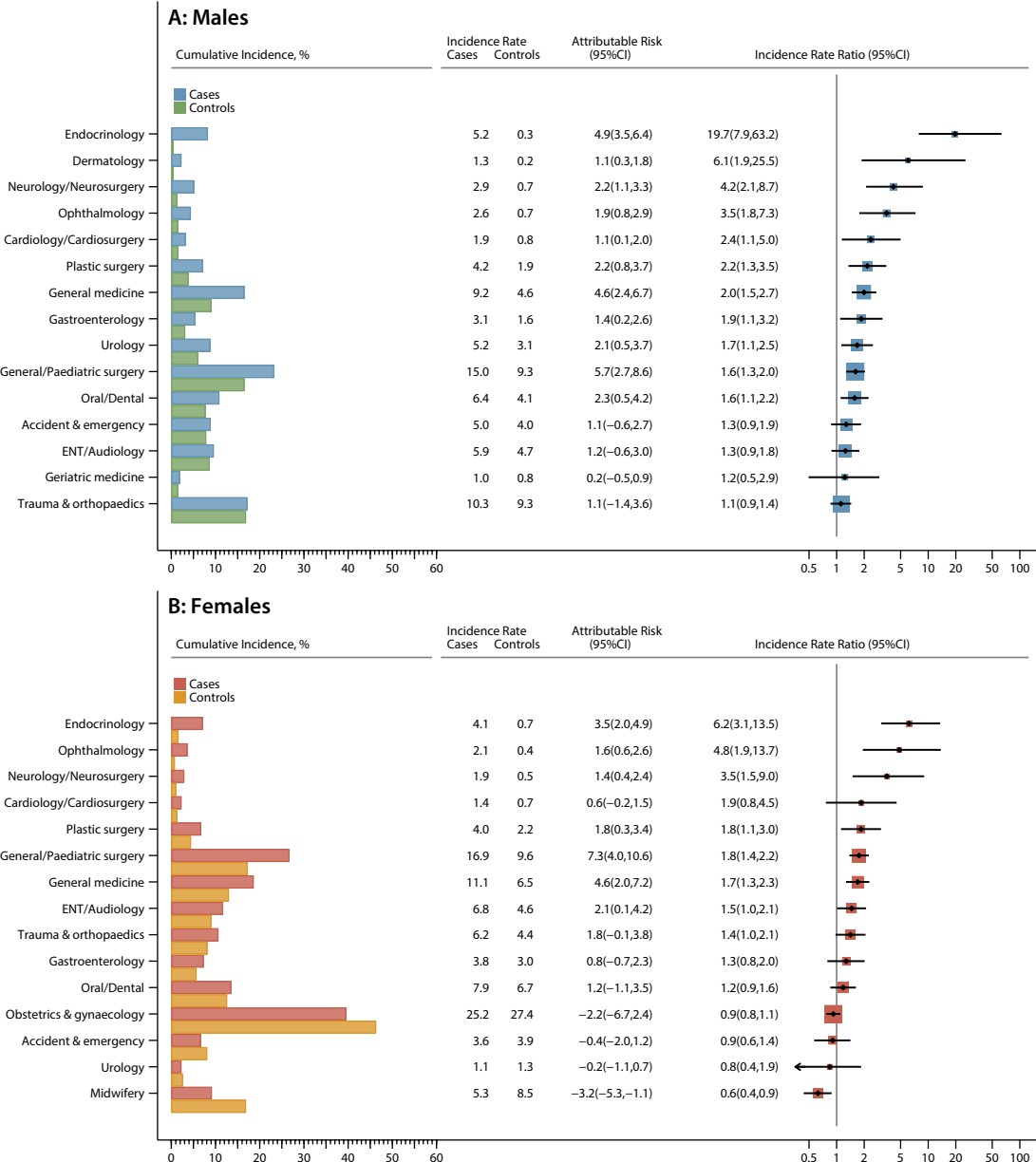

**Figure 4** Cumulative incidence (%), incidence rates (per 1000 person years), attributable risks and incidence rate ratios for the top 15 inpatient specialties (excluding paediatrics, haematology and oncology) with one or more admissions in the 5–25 years following diagnosis (cases diagnosed aged <15 years, 1992–1996) and their matched population controls. ENT, ear, nose and throat.

being hospitalised with endocrine or infectious diseases, as well as diseases involving the nervous, circulatory, skin, digestive, genitourinary, respiratory or musculoskeletal systems. With the information source mostly being inpatient records, reports using data from other healthcare settings are sparse[7 18] and not specific to survivors of ALL.[19 20] Importantly, while our findings are broadly consistent with these studies, our methods also highlight excesses in ophthalmology, ENT, and oral/dental specialties, as well as decreases in midwifery.

Ensuring homogeneity of treatment, the original study's diagnostic dates were contemporaneous with the national UKALL XI RCT, within which 90.5% of UKCCS 5-year survivors were enrolled.[22 30] This comparatively high-intensity RCT was the last UK childhood ALL trial prior to the stratification of patient treatment based on early bone marrow response to induction treatment (measured by percentage of leukaemia cells in the marrow).[23 31] Information on UKALL XI's scheduling of drugs is provided in the report by Hann *et al*,[22] several of which have been implicated, either singly or combined, in therapy-related effects (table 1). It is, however, important to note that several of the UKALL XI drugs are no longer used (eg, etoposide, 6-thioguanine) or are used more sparingly (eg, daunorubicin, cyclophosphamide, corticosteroids), and that cranial irradiation is no longer routinely administered, being reserved for infrequent particular indications.[32] Hence, the nature of

the UKCCS cohort means that it is well placed to provide baseline data against which to evaluate the potential impact of treatment de-escalation and other protocol modifications that occurred in later years. Other major strengths include its national coverage, completeness of case ascertainment, and large in-built sex-matched and age-matched general population comparator that is similarly distributed with respect to region and area-based deprivation (IMD)[25–27]; the latter obviating the need to rely on published national rates, as has mostly been the case in the UK thus far.[10 15 33–35]

With regard to weaknesses, the lack of data from other parts of the NHS, including primary care and adolescent psychosocial services, is an obvious deficiency that currently affects all UK record-linkage studies of the type described here. The relative paucity of information on psychological morbidities is particularly relevant to childhood cancer survivors, who are known to be at increased risk of a number of such disorders.[4 36 37] Missing information within the HES datasets is also an issue; diagnostic data, for example, are largely absent from outpatient HES. Furthermore, although steps to mitigate surveillance bias were taken by requiring at least two face-to-face outpatient visits to a given specialty, we cannot rule out the possibility that childhood cancer survivors were more likely to be referred to secondary care than their peers. With virtually complete linkage to existing national databases established, our future research will, however, continue to monitor the health of cancer survivors across their lifespan. As the cohort matures and the number of events increase, this will include more detailed examination of second malignancies, as well as factors potentially associated with ALL survival and/or outcome, such as socioeconomic status and treatment information (eg, risk strata, and relapse and/or transplant therapy).

Examining data in the broader context of symptomology based on ongoing medical referrals, this report focuses on the health of individuals who survived 5 years or more following a diagnosis of ALL as a child in the early 1990s; presenting an overview for all specialists who may come across unusual symptoms, signs and issues in those who had treatment for ALL at a young age. Future work to specifically address longer term health in relation to intensities and modifications of treatment including: central nervous system radiation (and dose), numbers of intensification blocks, particular drugs (eg, corticosteroids, cyclophosphamide, anthracyclines, etoposide, thioguanine) as well as early relapse and haematopoietic stem cell transplantation conditioning is planned. Thus far, adding to a large excess risk for death and cancer, we found that survivors experienced excess outpatient and inpatient activity across their TYA years, which was not related to routine follow-up monitoring. Involving most clinical specialties, the observed associations are striking; showing no signs of diminishing over the first 25 years of follow-up, the main exception is the comparatively low midwifery contact seen among female ALL survivors. All survivors of ALL are at increased risk of late effects

from treatment, and our findings underscore the need to take prior ALL diagnosis into account when interpreting seemingly unrelated symptoms in later life, which currently might not be recognised by specialties outside haemato-oncology. As importantly, there is also a need for provision of information to survivors themselves to encourage them to declare previous treatment for ALL as a child when they access healthcare.

**Contributors** ER, SK and JS contributed to the design of the original study; ER and EK instigated the present investigation and prepared the report. EK conducted the analyses, and all authors were involved in the development, reviewing, and approving the final manuscript. ER is the principal investigator and guarantor.

**Funding** This work was supported by Blood Cancer UK (grant number 15037) and Cancer Research UK (grant number 29685). AB is supported by a fellowship from the Fondation ARC pour la Recherche sur le Cancer (PDF20190508759).

**Competing interests** None declared.

**Patient consent for publication** Not applicable.

**Ethics approval** This study involves human participants and was approved by Yorkshire & the Humber-Leeds West Research Ethics Committee (reference 18/YH/0135). Exemption from Section 251 of the Health & Social Care Act (reference 18/CAG/0066)

**Provenance and peer review** Not commissioned; externally peer reviewed.

**Data availability statement** No data are available. Although ethical permissions and agreements with providers of national data mean that data deemed to have the potential to identify individuals cannot be transferred or accessed off site, UKCCS data are contributing to several ongoing research projects. For information on how to collaborate with UKCCS researchers and investigate questions of interest please email the corresponding author (ER).

**ORCID iDs**
Eleanor Kane http://orcid.org/0000-0002-7438-9982
Debra Howell http://orcid.org/0000-0002-7521-7402
Alexandra Smith http://orcid.org/0000-0002-1111-966X
Eve Roman http://orcid.org/0000-0001-7603-3704

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
