## [Reviewer comments · BMJ Open]

ARTICLE DETAILS

TITLE (PROVISIONAL)	Excess morbidity and mortality among survivors of childhood acute lymphoblastic leukaemia: 25 years of follow-up from the United Kingdom Childhood Cancer Study (UKCCS) population-based matched cohort
AUTHORS	Kane, Eleanor; Kinsey, Sally; Bonaventure, Audrey; Johnston, Tom; Simpson, Jill; Howell, Debra; Smith, Alexandra; Roman, Eve

VERSION 1 – REVIEW

REVIEWER	Dixon, Stephanie St Jude Children's Research Hospital, Oncology
REVIEW RETURNED	07-Oct-2021

GENERAL COMMENTS	The authors nicely present long-term mortality data from a population-based, case-control study of 5-year survivors of childhood ALL matched to non-cancer controls from the United Kingdom. This study adds to existing literature on late morbidity from childhood ALL by examining both inpatient and outpatient hospital/care encounters over a 25 year period among participants diagnosed 1992-1996. Although the authors nicely attempted to control for treatment by restricting participants to a 4 year period where the majority of pediatric ALL patients treated in the UK would have received UKALL XI protocol therapy (90% of 5-year survivors per this study report), a limitation is that they were not able to account for differing intensity of treatment within this cohort by risk-stratification of ALL, relapse status or receipt of bone marrow transplant. However, this is an important study of a large, population-based cohort of ALL survivors that identifies a significant increase in health care utilization compared to non-cancer controls in this relatively recently treated group of ALL survivors. Methods and Statistical Analyses: 1) Please clarify in the methods the definition of outpatient and inpatient "events", there is a statement in paragraph 3 of the results regarding two or more fact-to-face visits for outpatient encounter and this would be useful in the methods as well as a statement as to why that definition was chosen. Results: 2) The statement in the first paragraph of "Tracking children from around their 9th year of age through to their 29th" is confusing, this is the median of beginning follow-up and the median of end of follow-up. A better statement would be the median age at diagnosis and the median follow-up period or more clearly stating that survivors were followed from 5 years after diagnosis (median age 9.4 years, IQR XX) to death, 25 years from diagnosis or the end of the study period at a median age of 29.6 years, IQR XX.
---

	3) Please move information regarding methods of defining inpatient and outpatient encounters to the methods section and better define. Discussion: 4) Discussion regarding the lack of treatment information (or risk strata) as a limitation remains important. Despite using a short period of diagnosis to ensure most participants were treated on the same ALL protocol, some HR individuals on UKALL XI received 24 Gy CRT which would certainly impact morbidity including second cancer risk and endocrine risk. Additionally, although the authors note the appropriate references for treatment of UKALL XI some discussion of basic information would be helpful (EFS vs OS at 5 years as presumably a large proportion of those who had an event but survived received transplant and/or additional intensive chemotherapy). It is unclear based on this report if the high risk population who received CRT and relapse/transplant population from this era are driving these adverse events and increased care utilization. If this is the case, we would hope, as the authors suggest, significant reductions in the next generation of ALL survivors from the UK where upfront use of radiation therapy is not typically used and EFS has improved (therefore decreasing numbers of survivors who received relapse or transplant therapy).
--	---

REVIEWER	Lajous, Martin INSP, Center for Population Health Research
REVIEW RETURNED	22-Oct-2021

GENERAL COMMENTS	In this paper the authors compare mortality and morbidity throughout teenage and young adult years in childhood ALL survivors and controls from a national population-based case-control study in the UK the began recruitment in the 1992-1996 and was linked to deaths, cancer registration, hospital admission, and outpatient visits. Characterizing the magnitude of mortality and morbidity in childhood and teenage childhood cancer survivors may provide important insights on treatment-related sequelae. The data availability and completeness for the current analysis is impressive. However, excitement for this manuscript is significantly reduced by limitations on the description of the data, analytic choices and reporting, and interpretation. First, readers may be interested in understanding the validity of the cross-linkage to national databases and their completeness. Also, more detail on the types of hospital and outpatient visits are requires. Are all specialties included? What is the rationale for the categorization presented in the results? Second, far more details are required to understand the analytic strategy used. No information is provided on when follow-up time began for matched controls. We may assume that this was matched to the cases but it needs to be specified. The authors say the use “non-parametric time-to-event analyses”. Do they mean they estimated cumulative incidence and incidence rates? In the same sentence hazard ratios are mentioned these are derived from Cox proportional models. Were such models fit? Were any variables considered for adjustment? If not, why? Please specify and provide details on time scale and modeling assumptions. The authors mentioned that their cumulative incidence estimates take into account censoring. Please specify how this was done. Also, the reader can see in the results that the rates are per 1000 person-years this needs to be defined in the methods section. Please provide details on how attributable risks were calculated and what was the rationale for doing so. This is a causal concept so the authors need to articulate how a disease becomes an exposure of interest. Is it exposure to oncologic
---

	treatments that they have in mind? Third, some clarification needs to be made on presenting results. In Table 2 the authors present lineage for controls. Why? They also present cumulative mortality and then morality rate ratio. They should choose the appropriate incidence measure. If they are presenting rate ratios they should specify person-years and/or the mortality rate for cases and controls. Same choice should be made for report on cancer incidence. For clarity it may be easier for the reader if only malignancies and not cancer registrations are presented in the table. Finally, the discussion would require a more thorough quantitative comparison of results relative to the studies presented in Table1. Did the authors consider differences between cases and controls beyond age and sex. Why not adjust for factors that may be associated to ALL survival as well as outcomes in the study (e.g. region or SES)? Also, throughout the manuscript causal language is used (e.g. effect). This is inappropriate given the study design. Specific comments  1. While Table 1 may have been useful for writing this report I suggest making it part of the supplementary material. 2. Suggest revising the design in the abstract as this is a retrospective cohort.
--	---

VERSION 1 – AUTHOR RESPONSE

Reviewer: 1

Dr. Stephanie Dixon, St Jude Children's Research Hospital
Comments to the Author:

1.1 The authors nicely present long-term mortality data from a population-based, case-control study of 5-year survivors of childhood ALL matched to non-cancer controls from the United Kingdom. This study adds to existing literature on late morbidity from childhood ALL by examining both inpatient and outpatient hospital/care encounters over a 25 year period among participants diagnosed 1992-1996. Although the authors nicely attempted to control for treatment by restricting participants to a 4 year period where the majority of pediatric ALL patients treated in the UK would have received UKALL XI protocol therapy (90% of 5-year survivors per this study report), a limitation is that they were not able to account for differing intensity of treatment within this cohort by risk-stratification of ALL, relapse status or receipt of bone marrow transplant. However, this is an important study of a large, population-based cohort of ALL survivors that identifies a significant increase in health care utilization compared to non-cancer controls in this relatively recently treated group of ALL survivors.

Response: Thank you for these comments.

Methods and Statistical Analyses:

1.2 Please clarify in the methods the definition of outpatient and inpatient “events”, there is a statement in paragraph 3 of the results regarding two or more fact-to-face visits for outpatient encounter and this would be useful in the methods as well as a statement as to why that definition was chosen.

Response: Thank you for highlighting this omission – an issue that was also raised by Reviewer 2. The final paragraph of the Data and Statistical Analysis section has been modified and two references added. It now reads:

“Although the International Classification of Diseases (ICD) is used to code deaths and cancers, it is not used in outpatient HES, and the ICD codes in inpatient HES are not easily assigned to specific events. Each HES episode (inpatient and outpatient) is, however, assigned to an individual consultant whose clinical specialty categorization is recorded; and these categorizations were used in

the HES analyses presented here. Firstly, since childhood ALL survivors are routinely monitored via outpatient visits to paediatrics, haematology and/or oncology in the UK, HES attendances to these clinical specialties were separated from those to “other specialties” at the outset (Figure 2). All subsequent specialty-specific analyses presented in this report were restricted to the “other specialties” dataset. As in previous reports (28,29), a single inpatient admission and/or two or more specialty-specific face-to-face outpatient attendances were used to indicate potential health issues. Exact methods were used to plot estimates of the hospital visit hazard rates before calculating cumulative incidence (considering death as a competing risk), and risks associated with specific clinical specialties. All specialties were analysed, and findings for the 15 specialties with the highest cumulative incidences among the cases are presented. All analyses were conducted using Stata 16.1.”

Results:

1.3 The statement in the first paragraph of “Tracking children from around their 9th year of age through to their 29th” is confusing, this is the median of beginning follow-up and the median of end of follow-up. A better statement would be the median age at diagnosis and the median follow-up period or more clearly stating that survivors were followed from 5 years after diagnosis (median age 9.4 years, IQR XX) to death, 25 years from diagnosis or the end of the study period at a median age of 29.6 years, IQR XX.

Response: Thank you for this suggestion, which is much clearer. We have amended the first paragraph of the results accordingly, the relevant sentences now reading:

“Tracking children from 5 years after an ALL diagnosis to death or 25 years from diagnosis, this report’s primary focus is teenage and young adult (TYA) survivors; as they were aged around 9 years at the start of follow-up (median age 9.4 years, InterQuartile Range 8.0-12.1 years) and almost 30 by the end (median attained age 29.6 years, IQR 27.3-32.1). Between these ages, again as expected, the 5-year ALL survivors continued to experience more deaths (cumulative mortality 10.7%, 95% Confidence Interval 9.0-12.8) than their general population counterparts (cumulative mortality 0.6%, 95% CI 0.3-1.1), yielding a mortality rate ratio of 21.3 (95% CI 11.2-45.6); and were more likely to have a subsequent cancer registration (non-ALL malignancy Incidence Rate Ratio 9.9, 95% CI 4.1-29.1), although the numbers are small (32 versus 6).”

1.4 Please move information regarding methods of defining inpatient and outpatient encounters to the methods section and better define.

Response: Thank you - the definitions have been moved to the methods and clarification added (see response to section 1.2).

Discussion:

1.5 Discussion regarding the lack of treatment information (or risk strata) as a limitation remains important. Despite using a short period of diagnosis to ensure most participants were treated on the same ALL protocol, some HR individuals on UKALL XI received 24 Gy CRT which would certainly impact morbidity including second cancer risk and endocrine risk. Additionally, although the authors note the appropriate references for treatment of UKALL XI some discussion of basic information would be helpful (EFS vs OS at 5 years as presumably a large proportion of those who had an event but survived received transplant and/or additional intensive chemotherapy). It is unclear based on this report if the high risk population who received CRT and relapse/transplant population from this era are driving these adverse events and increased care utilization. If this is the case, we would hope, as the authors suggest, significant reductions in the next generation of ALL survivors from the UK where upfront use of radiation therapy is not typically used and EFS has improved (therefore decreasing numbers of survivors who received relapse or transplant therapy).

Response: We totally agree with the reviewer about the importance of treatment information, and have modified the Discussion in several places to further stress this aspect. As noted in Table 2, 90.5% of the 5-year survivors participated in a clinical trial (962 UKALL XI, and 17 infant ALL). The UKCCS can access UKALL XI, and has follow-up data from the trial to April 2007. As the reviewer knows, the main findings from this trial have already been published. Accordingly, using the ALL survivor cohort alone (the general-population comparison cohort is not required for such analyses), we plan to look more closely at the linked data and report the findings in a specialist

haematology/oncology journal. With this in mind the following text has been added to the Discussion:

“With virtually complete linkage to existing national databases established, our future research will, however, continue to monitor the health of cancer survivors across their life-span. As the cohort matures and the number of events increase, this will include more detailed examination of second malignancies, as well as factors potentially associated with ALL survival and/or outcome, such as socioeconomic status and treatment information (e.g. risk strata, and relapse and/or transplant therapy). “

All survivors of ALL are at increased risk of late effects from treatment, and the present report is concerned with the cohort of patients still alive 5 years from diagnosis, whether or not they relapsed. Focusing on the longer-term symptomatology/morbidity not (only) success in curing the initial ALL, it is aimed at the general clinical audience who may not be as familiar with the range of health problems associated with ALL therapies given many years earlier; and who are unlikely to have ready access to the treatments that were actually administered at the time. To emphasize this point, the last paragraph of the Discussion now reads:

“Examining data in the broader context of symptomology based on on-going medical referrals, the present report focuses on the health of individuals who survived five years or more following a diagnosis of ALL as a child in the early 1990s. Adding to a large excess risk for death and cancer, we found that survivors experienced excess outpatient and inpatient activity across their teenage and young adult years, which was not related to routine follow-up monitoring. Involving most clinical specialties, the observed associations are striking; showing no signs of diminishing over the first 25 years of follow-up, the main exception is the comparatively low midwifery contact seen among female ALL survivors. All survivors of ALL are at increased risk of late effects from treatment, and our findings underscore the need to take prior ALL diagnosis into account when interpreting seemingly unrelated symptoms in later life, which currently might not be recognized by specialties outside haemato-oncology. As importantly, there is also a need for provision of information to survivors themselves to encourage them to declare previous treatment for ALL as a child when they access healthcare. “

Reviewer: 2

Dr. Martin Lajous, INSP
Comments to the Author:

In this paper the authors compare mortality and morbidity throughout teenage and young adult years in childhood ALL survivors and controls from a national population-based case-control study in the UK the began recruitment in the 1992-1996 and was linked to deaths, cancer registration, hospital admission, and outpatient visits. Characterizing the magnitude of mortality and morbidity in childhood and teenage childhood cancer survivors may provide important insights on treatment-related sequelae. The data availability and completeness for the current analysis is impressive. However, excitement for this manuscript is significantly reduced by limitations on the description of the data, analytic choices and reporting, and interpretation.

Response: Thank you for raising these important points, and apologies for the lack of detail provided. The methods have now been reordered and more detail provided throughout.

2.1 First, readers may be interested in understanding the validity of the cross-linkage to national databases and their completeness.

Response: Thank you for raising this important point. We agree that these numbers would be useful to see, and have added the following text to the Data and statistical analysis section in the Methods. “Overall, 1574/1580 (99.6%) of the original cases and 2918/2923 (99.8%) of their first-choice controls were successfully traced and linked to national administrative databases.”

2.2 Also, more detail on the types of hospital and outpatient visits are requires. Are all specialties included?

Response: Again, apologies for not providing sufficient detail on this topic. Our study includes all hospital admissions (day and overnight) and all visits to outpatients; there are no exclusions, and

alspecialties are included. In the UK, childhood ALL survivors are monitored in haematology, oncology and/or paediatrics. So, in our first analyses (Figure 2), we separated these visits from those to “other specialties” (Figure 2). Similar analyses for inpatient admissions were conducted where rates were much lower (Supplementary Figure 1). Thereafter, to indicate which specialties survivors were visiting (as a proxy for the types of conditions- see below), we examined the number of cases and controls who had ever visited each specialty (2+ face-to-face outpatient visits or 1+ hospital admission); all specialties were analysed and findings are presented for the 15 specialties with the highest cumulative incidences among the cases.

We have added text to the methods to clarify what was included in the different analyses (see response to reviewer 1; section 1.2)

2.3 What is the rationale for the categorization presented in the results?

Response: Thank you for pointing out that more information about the categorization employed is required – an issue that was also raised by reviewer 1. This has been clarified in the Methods (see response to reviewer 1; section 1.2)

2.4 Second, far more details are required to understand the analytic strategy used. No information is provided on when follow-up time began for matched controls. We may assume that this was matched to the cases but it needs to be specified.

Response: This was an omission on our part, thank you for bringing it to our attention. The following text has now been added to the Methods:

“In the original case-control study, each unaffected control child was assigned a pseudo-diagnosis date that corresponded to their matched case’s exact age at diagnosis; and the dates of diagnosis/pseudo-diagnosis now mark the start of follow-up for the matched cohorts. “

2.5 The authors say the use “non-parametric time-to-event analyses”. Do they mean they estimated cumulative incidence and incidence rates?

Response: As rightly pointed out, this was confusing. Having deleted “non-parametric time-to-event analyses”, we hope it is now clear that exact methods were used to estimate cumulative incidence and incidence rates.

2.6 In the same sentence hazard ratios are mentioned these are derived from Cox proportional models. Were such models fit? Were any variables considered for adjustment? If not, why? Please specify and provide details on time scale and modeling assumptions.

Response: In the methods, we described using hazard rates (not ratios) which were estimated via exact methods with no adjustments; there was no modelling using Cox or other regressions. We employed the hazard rates to illustrate secondary care activity across follow-up in Figure 2 and Supplementary Figure 1; here standard Sata commands were used to plot estimates of the hazard function (hazard rate) via a weighted kernel-density estimate for the hazard contribution at each instance.

2.7 The authors mentioned that their cumulative incidence estimates take into account censoring. Please specify how this was done.

Response: Our apologies, we had omitted that cumulative incidences were calculated considering death as a competing risk; this has now been added to the methods.

2.8 Also, the reader can see in the results that the rates are per 1000 person-years this needs to be defined in the methods section.

Response: The results include rates on two different scales: hazard rates per year for all hospital visits (Figure 2); and incidence rates per 1000 person-years in the analyses of ever visiting a specialty (Figures 3 and 4). We opted for the former scale to give a measure of how often a childhood ALL survivor might expect to visit hospital in a year. The purpose of presenting incidence

rates was to aid derivation of one of our other main risk estimates -the incidence rate ratios; the incidence rates were scaled by 1000 for readability since these are a measure of ever visiting (rather than all visits). To avoid confusion, we have clearly labelled the different rates in the figure titles.

2.9 Please provide details on how attributable risks were calculated and what was the rationale for doing so. This is a causal concept so the authors need to articulate how a disease becomes an exposure of interest. Is it exposure to oncologic treatments that they have in mind?

Response: Apologies again for any lack of clarity. Attributable risks (also known as absolute risk increases /risk differences) were calculated in the standard way (rate in the childhood ALL survivor's cohort - rate in the age- sex matched comparator cohort), allowing our findings to be compared to those presented in other reports. As noted in the introduction and discussion, these differences may be due to ALL treatments. In response to Reviewer 1's comments (section 1.5), the discussion on this topic has been extended.

2.10 Third, some clarification needs to be made on presenting results. In Table 2 the authors present lineage for controls. Why?

Response: Our apologies, this is an error; the number of controls by their matched cases' lineage has been removed from the table.

2.11 They also present cumulative mortality and then morality rate ratio. They should choose the appropriate incidence measure.

Response: We presented these two measures to be comparable to the literature; the cumulative mortality measure in this type of study being easier to convey- as a corollary to survival- than a mortality rate. However, in order that mortality rates (and so the mortality rate ratio) can be derived, we have added the person-years of follow-up to Table 2.

2.12 If they are presenting rate ratios they should specify person-years and/or the mortality rate for cases and controls. Same choice should be made for report on cancer incidence. For clarity it may be easier for the reader if only malignancies and not cancer registrations are presented in the table.

Response: In response to the point above and to that made here, we have added person-years of follow-up to Table 2. We chose to report both total cancer registrations and malignancies – as has been done in other reports of childhood cancer survivors- because not all neoplasms are malignant.

2.13 Finally, the discussion would require a more thorough quantitative comparison of results relative to the studies presented in Table1.

Response: We would have liked to have compared the results in a more quantitative manner to those of other reports, but ours is the only study to show outpatient activity, from early survivorship through to young adult years, and to report findings in the context of an unaffected general population comparator. Across the survivorship studies, there is considerable variation in design- including, for example, the settings, case/treatment era, comparators used, markers of morbidities, and length of follow-up; all of which may affect the size of the risk estimates. Nevertheless, associations are generally large, and we see some consistency in the types of morbidities survivors might experience. Hence, for these reasons, we opted instead to present the main findings from previous studies in the last column of Table 1. With two new references having been published in recent months, we have also taken this opportunity to add these studies to the Table (Jensen 2021- reference number 7; and Chang 2021- reference number 20).

2.14 Did the authors consider differences between cases and controls beyond age and sex. Why not adjust for factors that may be associated to ALL survival as well as outcomes in the study (e.g. region or SES)?

Response: Thank you for this comment. Obviously, like the reviewer, we are well aware that many other factors could impact on outcome. As our study includes all cases of ALL diagnosed in England during the early 1990s, our findings illustrate the true landscape of hospital activity among this group

of survivors – which is the purpose of the present report. As expected, given the design of the original study, the cases and their age- and sex-matched first-choice controls had similar regional and Index of Multiple Deprivation (IMD – income domain) distributions at the time of diagnosis (and also at birth). Given this, it is not surprising that no IMD differences between the groups were apparent 5 years after diagnosis/pseudo-diagnosis – which is now stated more explicitly in the paper (see the last sentence of the third paragraph of the discussion). We will, of course, report differences, should they emerge in the future, and a sentence on possible future work has been added to the discussion (last sentence, fourth paragraph of the discussion).

2.15 Also, throughout the manuscript causal language is used (e.g. effect). This is inappropriate given the study design.

Response: Thank you- we have changed the causal language as appropriate.

Specific comments

1. While Table 1 may have been useful for writing this report I suggest making it part of the supplementary material.

Response: Table 1 is shown in the main paper to place our study within the context of the many studies on childhood leukaemia survivors. We feel the table illustrates our study's novelty, and as stated above (see section 2.13), it also serves to provide a summary of the main findings of the other studies.

2. Suggest revising the design in the abstract as this is a retrospective cohort.

Response: To clarify, while we accessed the data retrospectively, data from the administrative healthcare databases are recorded contemporaneously by bodies such as the National Health Service. The design is prospective not historical, and this has now been clarified in the methods

VERSION 2 – REVIEW

REVIEWER	Dixon, Stephanie St Jude Children's Research Hospital, Oncology
REVIEW RETURNED	13-Jan-2022

GENERAL COMMENTS	The authors have made a number of suggested revisions to the previously reviewed manuscript which have improved this version. This study presents long-term mortality data from a population-based, case-control study of 5-year survivors of childhood ALL matched to non-cancer controls from the United Kingdom and adds to existing literature on late morbidity from childhood ALL by examining both inpatient and outpatient hospital/care encounters over a 25 year period among participants diagnosed 1992-1996. A few minor concerns remain: The authors report results of incidence rate ratios and mortality rate ratios in the results but do not discuss these estimates in the methods. When analyses were conducted was there any adjustment for potential confounders? Was it felt that the study design and control selection adequately controlled for confounding socio-demographic variables? 2) The discussion has been strengthened by the additions. The results and discussion would be further strengthened if information on cranial radiation (either upfront or for later treatment of relapse) or receipt of additional intense treatment for relapse, including transplant. It is possible that a small subset of survivors who received cranial radiation or transplant are driving a portion of the
---

	increased health-care utilization identified in this study. If this information is not available it may warrant mention of this possibility in the discussion.
--	--

VERSION 2 – AUTHOR RESPONSE

Reviewer: 1

Dr. Stephanie Dixon, St Jude Children's Research Hospital

Comments to the Author:

The authors have made a number of suggested revisions to the previously reviewed manuscript which have improved this version. This study presents long-term mortality data from a population-based, case-control study of 5-year survivors of childhood ALL matched to non-cancer controls from the United Kingdom and adds to existing literature on late morbidity from childhood ALL by examining both inpatient and outpatient hospital/care encounters over a 25 year period among participants diagnosed 1992-1996.

A few minor concerns remain:

The authors report results of incidence rate ratios and mortality rate ratios in the results but do not discuss these estimates in the methods. When analyses were conducted was there any adjustment for potential confounders? Was it felt that the study design and control selection adequately controlled for confounding socio-demographic variables?

Please see response to editor's comments above.

2) The discussion has been strengthened by the additions. The results and discussion would be further strengthened if information on cranial radiation (either upfront or for later treatment of relapse) or receipt of additional intense treatment for relapse, including transplant. It is possible that a small subset of survivors who received cranial radiation or transplant are driving a portion of the increased health-care utilization identified in this study. If this information is not available it may warrant mention of this possibility in the discussion.

Response: This paper presents an overview for all specialists who may come across unusual symptoms, signs and issues in those having had treatment for ALL at a young age. As such, our aim is to reach non-haematology-oncology practitioners who may be meeting patients for the first time many years later, rather than a Haem/Onc audience who would value more treatment detail. Hence, the findings presented relate to the wide-ranging associations observed across the whole population of ALL survivors.

Importantly, the background literature and analytical methods needed to examine treatment effects differ from those used in the present report which uses a general population comparator. Accordingly, as detailed in the report, our intention is a further publication for a more specialist journal to specifically address intensities and modifications of treatment including: CNS radiation (and dose), numbers of intensification blocks, particular drugs (e.g. corticosteroids, cyclophosphamide, anthracyclines, etoposide, thioguanine) as well as early relapse and HSCT conditioning. Interestingly, as the reviewer knows, although most relapses in childhood ALL are seen in the first 5 years from diagnosis, there will also be a minority of patients needing treatment for relapse +/- HSCT after 5 years from end of treatment; and in this context looking at longer-term associations is clearly worthwhile. As requested, to sign-post this we have added text to the last paragraph of the Discussion as follows:

“Examining data in the broader context of symptomology based on on-going medical referrals, this report focuses on the health of individuals who survived five years or more following a diagnosis of ALL as a child in the early 1990s; presenting an overview for all specialists who may come across unusual symptoms, signs and issues in those having had treatment for ALL at a young age. Future work to specifically address longer-term health in relation to intensities and modifications of treatment including: central nervous system radiation (and dose), numbers of intensification blocks, particular drugs (e.g. corticosteroids, cyclophosphamide, anthracyclines, etoposide, thioguanine) as well as early relapse and haematopoietic stem cell transplantation conditioning is planned.”